# Differentially expressed genes reflect disease-induced rather than disease-causing changes in the transcriptome

Eleonora Porcu [1,2,3✉], Marie C. Sadler [2,3], Kaido Lepik[4,5], Chiara Auwerx [1,2,3], Andrew R. Wood [6], Antoine Weihs [7], Maroun S. Bou Sleiman[8], Diogo M. Ribeiro [2,9], Stefania Bandinelli[10], Toshiko Tanaka[11], Matthias Nauck [12,13], Uwe Völker [13,14], Olivier Delaneau [2,9], Andres Metspalu [15], Alexander Teumer [13,16], Timothy Frayling [17], Federico A. Santoni[18], Alexandre Reymond[1] & Zoltán Kutalik [2,3,6,9]

Comparing transcript levels between healthy and diseased individuals allows the identification of differentially expressed genes, which may be causes, consequences or mere correlates of the disease under scrutiny. We propose a method to decompose the observational correlation between gene expression and phenotypes driven by confounders, forward- and reverse causal effects. The bi-directional causal effects between gene expression and complex traits are obtained by Mendelian Randomization integrating summary-level data from GWAS and whole-blood eQTLs. Applying this approach to complex traits reveals that forward effects have negligible contribution. For example, BMI- and triglycerides-gene expression correlation coefficients robustly correlate with trait-to-expression causal effects ($r_{BMI} = 0.11$, $P_{BMI} = 2.0 \times 10^{-51}$ and $r_{TG} = 0.13$, $P_{TG} = 1.1 \times 10^{-68}$), but not detectably with expression-to-trait effects. Our results demonstrate that studies comparing the transcriptome of diseased and healthy subjects are more prone to reveal disease-induced gene expression changes rather than disease causing ones.

[1] Center for Integrative Genomics, University of Lausanne, Lausanne, Switzerland. [2] Swiss Institute of Bioinformatics, Lausanne, Switzerland. [3] University Center for Primary Care and Public Health, Lausanne, Switzerland. [4] Institute of Computer Science, University of Tartu, Tartu, Estonia. [5] Estonian Genome Centre, Institute of Genomics, University of Tartu, Tartu, Estonia. [6] Genetics of Complex Traits, College of Medicine and Health, University of Exeter, Exeter, Devon, UK. [7] Department of Psychiatry and Psychotherapy, University Medicine Greifswald, Greifswald, Germany. [8] Laboratory of Integrative Systems Physiology, Institute of Bioengineering, Ecole Polytechnique Fédérale de Lausanne, Lausanne 1015, Switzerland. [9] Department of Computational Biology, University of Lausanne, Lausanne, Switzerland. [10] Local Health Unit Toscana Centro, Florence, Italy. [11] Clinical Res Branch, National Institute of Aging, Baltimore, MD, USA. [12] Institute of Clinical Chemistry and Laboratory Medicine, University Medicine Greifswald, Greifswald, Germany. [13] DZHK (German Centre for Cardiovascular Research), partner site Greifswald, Greifswald, Germany. [14] Interfaculty Institute for Genetics and Functional Genomics, University Medicine Greifswald, Greifswald, Germany. [15] Estonian Biobank, University of Tartu, Tartu, Estonia. [16] Institute for Community Medicine, University Medicine Greifswald, Greifswald, Germany. [17] University of Exeter Medical School, University of Exeter, Exeter Devon, UK. [18] Endocrine, Diabetes, and Metabolism Service, Lausanne University Hospital, Lausanne, Switzerland. ✉email: eleonora.porcu@unil.ch

To unravel the genetics of complex diseases and traits causes, multiple approaches have concentrated on contrasting the expression of mRNA transcripts in two different groups of samples to understand how genes are expressed in health and disease[1–4]. This allows identifying differentially expressed genes (DEGs) that can be used to obtain mechanistic insights from diseases or serve as clinical biomarkers for early diagnostics. However, DEG analyses are unable to distinguish between causes, consequences, or mere correlations between gene expression and phenotypes. To understand the contributions to observed trait-expression correlations, both the assessment of bidirectional causal effects and the impact of (unmeasured) confounders are needed. We argue that if the observed correlations and bidirectional causal effects are estimated, the contribution of such confounders can be evaluated.

Genome-wide association studies (GWAS) identified thousands of common genetic variants associated with complex human traits[5] and studies on expression quantitative trait loci (eQTLs) showed how genetic variants contribute to the regulation of gene expression levels[6]. The overlay of the two methodologies showed that trait-associated SNPs are three times more likely to be eQTLs[7–10], suggesting that gene expression is a reliable intermediary between DNA variation and higher-order complex phenotypes. Starting from this hypothesis, many statistical approaches integrating GWAS and eQTLs summary statistics have been proposed to detect these overlapping associations[9,11,12]. However, while these studies aim to identify genes whose (genetically determined) expression is significantly associated with complex traits, they do not aim to estimate the strength of the causal effect and are unable to distinguish causation from pleiotropy (i.e., when a genetic variant independently affects gene expression and phenotype). This challenge can be addressed by combining summary-level data from eQTL and GWAS studies in a two-sample Mendelian Randomization framework[13] to evaluate whether gene expression has a causal influence on a complex trait. Such methods successfully identified thousands of genes associated with complex traits.

Yet, these transcriptome-wide approaches only use cis-eQTLs as instruments to tease out the causal effect of gene expression on a complex trait even though the variation in gene expression may be secondary to, rather than causal for, the disease process ("reverse causation"). Disease-associated genetic variants affect expression levels more often in trans than in cis[14]. Hence, polygenic risk scores (PRS) have been used to evaluate the association between genetically predicted complex traits and gene expression levels[14]. However, PRS-based approaches are prone to detect associations merely due to pleiotropic SNPs.

In this work, to circumvent this issue and elucidate the impact of diseases on the transcriptome program at a large scale and in a principled way, we propose a reverse transcriptome-wide Mendelian randomization approach (revTWMR), which integrates summary-level data from GWAS and trans-eQTLs studies in an MR framework to estimate the causal effect of phenotypes on gene expression. By combining revTWMR results with the causal effects of gene expression on phenotypes—estimated by transcriptome-wide Mendelian randomization (TWMR)[15]—we obtain a clear picture of the bidirectional causal effects between gene expression and complex traits (Fig. 1) and evaluate their contribution to their observational correlation.

## Results

**Overview of the approach.** We recently developed a transcriptome-wide summary statistics-based Mendelian randomization approach (TWMR[15]) integrating summary-level data from GWAS and cis-eQTL studies. Applying TWMR to summary data from whole blood cis-eQTL meta-analyses from >32,000 individuals (eQTLGen Consortium[14]) and publicly available GWAS summary statistics revealed an atlas of putative functionally relevant genes for several complex human traits[15]. This approach can be reversed to design a multi-instrument MR approach to estimate the causal effect of a phenotype (exposure) on gene expression (outcome) (revTWMR, Fig. 1). For each gene, using the inverse-variance weighted meta-analysis of ratio estimates from summary statistics[16], we estimate the causal effect of a phenotype on the expression of the probed gene as

$$\hat{\alpha} = \frac{\sum_{j=1}^{N} \beta_j \gamma_j}{\sum_{j=1}^{N} \beta_j^2} \qquad (1)$$

where $\beta_j$ and $\gamma_j$ are the standardized effect sizes of $SNP_j$ on the phenotype and on the expression level of the probed gene, respectively, and $N$ is the number of independent SNPs used as instrumental variables.

**Applying revTWMR to GWAS and eQTL summary statistics.** We applied revTWMR to assess causal associations between 12 complex traits—body mass index (BMI), Crohn's disease (CD), educational attainment (EDU), fasting glucose (FG), high-density lipoprotein (HDL), height, low-density lipoprotein (LDL), rheumatoid arthritis (RA), schizophrenia (SCZ), total cholesterol (TC), triglycerides (TG), and waist-to-hip ratio adjusted for BMI (WHRadjBMI)—and the expression of 19,942 genes. We combined summary whole blood trans-eQTLs data from the eQTL-Gen Consortium[14], with large publicly available GWAS for the traits of interest[17–22] (see Methods). Together, we identified 46 genes significantly affected by at least one phenotype ($P_{revTWMR} < 2.5 \times 10^{-6} = 0.05/19,942$), often corroborating known biological associations (Supplementary Data 1). In parallel, we performed TWMR analyses on the same set of traits, allowing testing for the presence of bidirectional effects (see Methods) (Supplementary Data 2).

The most influential traits in our analysis were TG and RA, significantly influencing the expression of 26 and 15 genes, respectively. These were analyzed for functional enrichment with UniProtKB[23], KEGG pathway[24], Gene Ontology[25], and InterPro[26]. For TG, cholesterol metabolism (UniProt KW-0153)

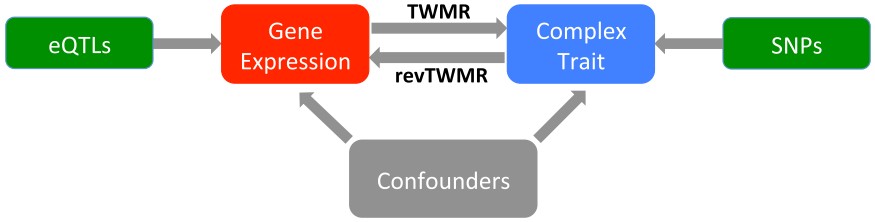

**Fig. 1 TWMR and revTWMR.** Schematic representation of how TWMR and revTWMR dissect bidirectional causal and confounder contributions to the observed correlation between gene expression and phenotype.

was the most significantly enriched class (Supplementary Data 3). For RA, immunoglobulin-related terms (InterPro IPR013106, IPR007110, and IPR013783) were the top significantly enriched classes (Supplementary Data 4). Closer investigation revealed that this enrichment is due to the presence of 7 T-cell receptor α and β variable genes (*TRAV* and *TRBV*). Interestingly, a bias in Vβ gene utilization by T cells in patients suffering from RA was reported[27].

Focusing on serum lipid levels (Supplementary Data 5), revTWMR revealed that in addition to the 26 genes affected by TG, the expression of eight genes is altered by HDL-cholesterol levels. In line with the commonly observed negative correlation between HDL and TG[28], five genes were impacted by both traits with an opposite direction of the causal effect (Supplementary Data 1). Regarding the impact of high HDL levels, we found that it reduced the expression of squalene synthase (*FDFT1*; $\alpha_{\text{revTWMR}} = -0.14$, $P_{\text{revTWMR}} = 1.3 \times 10^{-10}$, Supplementary Fig. 1), a key enzyme of the cholesterol biosynthesis pathway[29]. Interestingly, serum levels of squalene, the product of squalene synthase were found to negatively correlate with HDL-cholesterol[30]. Genes involved in cholesterol transport were impacted too: high HDL negatively impacted the expression of the LDL receptor (*LDLR*; $\alpha_{\text{revTWMR}} = -0.11$, $P_{\text{revTWMR}} = 1.0 \times 10^{-06}$, Supplementary Fig. 1), while having a positive impact on *MYLIP* (also known as *IDOL*; $\alpha_{\text{revTWMR}} = 0.18$, $P_{\text{revTWMR}} = 2.2 \times 10^{-16}$, Supplementary Fig. 1), a ubiquitin ligase that induces degradation of the LDL receptor[31]. In parallel, HDL increased the expression of the *ABCA1* ($\alpha_{\text{revTWMR}} = 0.24$, $P_{\text{revTWMR}} = 9.5 \times 10^{-29}$, Supplementary Fig. 1) and *ABCG1* ($\alpha_{\text{revTWMR}} = 0.19$, $P_{\text{revTWMR}} = 4.1 \times 10^{-20}$, Supplementary Fig. 1), two transporters responsible for cholesterol efflux from macrophages[32]. While we did not observe a significant effect of *ABCA1* on HDL and TC levels through TWMR, an association between *ABCA1* and these two traits was previously reported by a GWAS[33], suggesting a complex regulatory mechanism. Together, these results are reminiscent of the well-described negative feedback mechanisms that tightly control cholesterol biosynthesis and uptake[34].

As the other traits influenced only a smaller number of genes, no further significant enrichments were found. Nevertheless, a gene-by-gene investigation revealed many known or highly plausible associations, such as the significant effect of BMI on *ALDH1A1* ($\alpha_{\text{revTWMR}} = -0.17$, $P_{\text{revTWMR}} = 2.2 \times 10^{-06}$, Supplementary Fig. 1), an enzyme that converts retinaldehyde to retinoic acid[35]. Retinoids have long been implicated in adipogenesis[36,37] and *ALDH1A1* expression in visceral adipose tissue was shown to positively correlate with BMI[38].

Despite strong indications of functional relevance, most revTWMR-implicated genes fall into genomic regions completely missed by GWAS, as is illustrated by the fact that revTWMR p values are completely uncorrelated ($r < 0.05$) with those obtained by classical gene-based GWAS test performed using PASCAL[39] (See Methods; Supplementary Fig. 2). In line with this observation, only one out of the 46 revTWMR-identified genes were significant for TWMR: *FDFT1* shows a negative causal feedback loop between its expression and TG ($\alpha_{\text{TWMR}} = -0.04$, $P_{\text{TWMR}} = 1.6 \times 10^{-31}$ and $\alpha_{\text{revTWMR}} = 0.15$, $P_{\text{revTWMR}} = 1.3 \times 10^{-09}$, Supplementary Fig. 1).

To test the robustness of revTWMR, we performed MR analysis using two alternative approaches allowing the presence of invalid instruments: a weighted median method that assumes that a majority of genetic variants are valid instruments[40] and a weighted mode-based estimation method that assumes a plurality of genetic variants are valid instruments[41]. Results strongly supported the robustness of the IVW-based findings as 47 out of the 51 trait-gene revTWMR associations were significant in at least one of these methods ($P_{\text{MR}} < 0.05/51$; Supplementary Data 6). This analysis revealed 32 additional genes significantly

affected by RA (30 genes), TC (1), and TG (1) (Supplementary Data 7). Of note, the additional 30 genes associated with RA, strengthened the previously detected enrichment for the immunoglobulin InterPro functional groups (Supplementary Data 8).

**Pleiotropic SNPs lead to biased causal effect estimates**. The validity of revTWMR, as any MR approach, relies on three assumptions about the instruments: (i) they must be sufficiently strongly associated with the exposure; (ii) they should not be associated with any confounder of the exposure-outcome relationship; and (iii) they should be associated with the outcome only through the exposure. The third assumption is crucial as MR causal estimates will be biased in the presence of pleiotropy[42,43]. Accordingly, revTWMR assumes that all genetic variants used as instrumental variables affect the gene expression only through the phenotype under scrutiny and not through independent biological pathways.

To test for the presence of pleiotropy, we used a similar approach to MR-PRESSO global test[43,44], performing Cochran's Q test. Under the assumption that the majority of SNPs influence gene expression only through the phenotype tested in the model, SNPs violating the third MR assumption would significantly increase the Cochran's heterogeneity Q statistic (see Methods), allowing their detection and exclusion. This was the case for 16 of the 52 originally significant trait → gene associations. Out of these 16 associations, nine passed the heterogeneity test after removing pleiotropic SNPs from the instrumental variables. Moreover, this procedure led to the identification of six additional associations initially masked by heterogeneity, bringing the final number of robust associations to 51 (Supplementary Data 1). Importantly, revTWMR, like other MR methods, discriminates likely causal effects from pleiotropy, as illustrated by the example of *STX1B*, a gene that was found to be associated with EDU through a PRS approach ($P_{\text{PRS}} = 1.3 \times 10^{-20}$)[14]. Applying revTWMR, we did not observe an association between EDU and *STX1B* ($\alpha_{\text{revTWMR}} = 0.03$, $P_{\text{revTWMR}} = 0.83$) and detected a highly pleiotropic variant, rs2456973, strongly associated with hematological and anthropometric traits[45] (Supplementary Data 9 and Supplementary Fig. 3).

**Trait correlation**. Exploring the shared effect of complex traits and diseases on transcriptional programs can provide useful etiological insights. Hence, for every phenotype-pair ($P_i$, $P_j$) we computed the gene expression perturbation correlation between the respective causal effect estimates of each phenotype on the gene expression ($\hat{\rho}_{P(i,j)} = \text{corr}(\alpha_{Pi \rightarrow E}, \alpha_{Pj \rightarrow E})$) across a subset of 2974 independent genes across the genome[15]. Among the 55 pairs of traits, we found 21 significant correlations (FDR <1%). We compared these results with the genetic correlation ($\hat{\rho}_G$) between traits estimated by LD score regression[46] and found a remarkable concordance between the two estimates ($r = 0.84$). On average, $\hat{\rho}_P$ represents 56% of $\hat{\rho}_G$. Although $\hat{\rho}_G$ having smaller variance may explain part of this attenuation, we think that the main reason behind this observation is that only a part of $\hat{\rho}_G$ translates into consequences on gene expression level in whole blood (Supplementary Fig. 4). In particular, nine pairs of traits showed significance for both $\hat{\rho}_P$ and $\hat{\rho}_G$, whereas 12 were significant only for $\hat{\rho}_P$, and seven only for $\hat{\rho}_G$. Among the significant correlations not identified by LD score regression $\hat{\rho}_G$, we found that HDL and LDL are negatively correlated ($\hat{\rho}_P = -0.13$, FDR $= 3.1 \times 10^{-09}$) and that RA positively correlated with several traits: CD ($\hat{\rho}_P = 0.08$, FDR $= 4.0 \times 10^{-04}$), SCZ ($\hat{\rho}_P = 0.14$, FDR $= 1.5 \times 10^{-10}$), height ($\hat{\rho}_P = 0.09$, FDR $= 6.0 \times 10^{-05}$), TC ($\hat{\rho}_P = 0.08$, FDR $= 4.0 \times 10^{-04}$), and TG ($\hat{\rho}_P = 0.12$, FDR $= 4.8 \times 10^{-08}$) (Supplementary Data 10).

**Partitioning the observational correlation**. As a proof-of-concept, we asked how highly revTWMR-identified causal genes would rank in a DEG analysis. To address this question, we collected the observational correlation estimates between whole blood gene expression levels and the quantitative traits in three independent European cohorts (EGCUT ($N = 488$), InChianti ($N = 609$), and SHIP-Trend ($N = 991$)).

Correlating revTWMR effects to observational correlations (equivalent to DEG analysis), we found a significant agreement for all the traits (Table 1). We reestimated these correlations accounting for the error in the compared estimates (regression dilution bias) (see Methods). No significant correlation between observational correlations and the causal effects of the gene expression on phenotypes estimated by TWMR was observed (Table 1). Of note, when we correlated the $P$ values of the observational correlations with those obtained by conventional gene-based tests using GWAS results, we detected a significant concordance only for HDL ($r = 0.05$, $P = 1.3 \times 10^{-10}$) and TG ($r = 0.03$, $P = 5.7 \times 10^{-04}$) (Supplementary Data 11).

As we previously showed that causal feedback loops are rare (i.e., $\alpha_{TWMR} * \alpha_{revTWMR} = 0$), the observational correlation ($r$) can be approximated as the sum of the bidirectional effects estimated by TWMR and revTWMR plus the contribution of the

confounding factors (see Methods). Hence, we calculated the proportion of correlation due to confounders. For each gene we calculated the contribution of TWMR and revTWMR as $\frac{\alpha_{TWMR}}{r}$ and $\frac{\alpha_{revTWMR}}{r}$, respectively. Consequently, the contribution of confounders is $1 - \frac{\alpha_{TWMR}}{r} - \frac{\alpha_{revTWMR}}{r}$. In each correlation bin (Fig. 2) we combined such contributions using inverse-variance meta-analysis and revealed that the observed correlation between gene expression and phenotype is mainly driven by confounders. For example, for genes correlated ($|r| > 0.1$) with BMI, 83% ($P < 5.0 \times 10^{-324}$) of the correlation is due to the confounders, 17% ($P = 6.7 \times 10^{-45}$) to the effect of BMI on gene expression and 0% ($P = 0.67$) to the forward effect (Fig. 2 and Supplementary Data 12). A similar scenario was observed for TG: 90% ($P < 5.0 \times 10^{-324}$) of the correlation is due to confounders and only 10% ($P = 2.9 \times 10^{-35}$) and 0% ($P = 0.98$) are due to reverse and forward effect of the gene expression on TG, respectively. For HDL we observed a stronger effect due to confounders (94%, $P < 5.0 \times 10^{-324}$) and a mild reverse effect (6%, $P = 3.9 \times 10^{-15}$) (Fig. 2).

**Genes affected by lipid traits are linked to drug targets**. It is important to note that since GWAS findings point to loci underlying disease susceptibility, changes in expression detected by revTWMR do not necessarily represent the consequence of the disease but can also reflect the consequences of a genetic predisposition to that disease. Therefore, identified genes might represent early biomarkers of disease (predisposition) and modulation of their expression could be a promising therapeutical strategy. For this reason, we assessed whether the protein products of the transcripts identified by our revTWMR analysis are targets of drugs used to treat the disease in question. We started by defining a set of drugs relevant to the traits under investigation according to DrugBank[13]. Next, we retrieved high confidence interactions (confidence score >0.7) involving these drugs, from STITCH, a manually curated database of predicted and experimental chemical–protein interactions[12]. We then searched for proteins that were (a) identified as dysregulated by revTWMR and (b) targeted by a drug indicated for the treatment of a given trait.

The gene product of 4 out of the 8 genes detected by revTWMR for HDL-cholesterol met these criteria: phospholipid-transporting ATPase ABCA1 (*ABCA1*), squalene synthase (*FDFT1*), low-density lipoprotein receptor (*LDLR*), and sterol regulatory element-binding protein 1 (*SREBF1*) which interact with atorvastatin, lovastatin, pravastatin, and simvastatin. We

---

**Table 1 Correlation between observational phenotype-gene expression correlation and revTWMR and TWMR effects.**

| Trait | revTWMR | | TWMR | |
|---|---|---|---|---|
| | correlation (adjusted) | *P* value | correlation | *P* value |
| BMI | 0.11 (0.37) | 1.97E-51 | 0 | 0.75 |
| EDU | 0.04 (0.29) | 1.50E-08 | 0.02 | 0.04 |
| Fasting glucose | 0.08 (0.24) | 2.56E-17 | 0.01 | 0.40 |
| HDL | 0.10 (0.27) | 1.86E-43 | 0.01 | 0.18 |
| height | 0.09 (0.38) | 3.26E-37 | 0.01 | 0.33 |
| LDL | 0.02 (0.09) | 5.36E-04 | 0.02 | 0.05 |
| TC | 0.04 (0.13) | 5.30E-08 | 0.03 | 0.01 |
| TG | 0.13 (0.32) | 1.11E-68 | 0 | 0.73 |
| WHR | 0.02 (0.14) | 1.74E-04 | 0.01 | 0.40 |

For each phenotype available in at least two cohorts, we calculated the correlation between the observational correlation estimates and the revTWMR and TWMR effects. The $P$ value indicates the significance of the correlation coefficient calculated using a two-sided $t$-test. For significant correlations, we computed the adjusted correlation correcting for regression dilution bias.

---

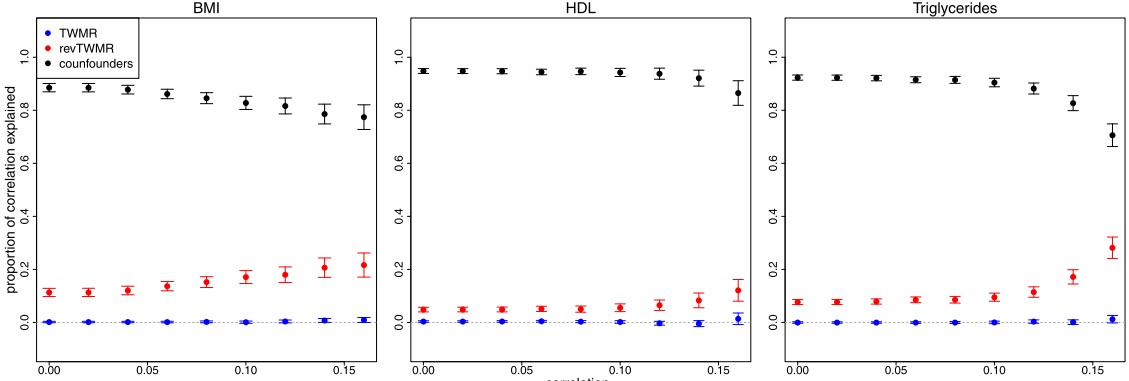

**Fig. 2 Partitioning the gene expression-trait observational correlation for BMI, HDL, and triglycerides.** Using all the genes tested by TWMR and revTWMR ($N = 10,395$ for BMI, $N = 10,391$ for HDL, and $N = 10,390$ for TG), for each bin of correlation (absolute value) we plotted the combined contributions of the forward (TWMR, blue dots) and reverse (revTWMR, red dots) effect of the gene expression on the trait, the contribution of confounders (black dots). Data were presented as estimated contributions and 95% confidence intervals.

found that *ABCA1*, *FDFT1*, and *SREBF1* are also dysregulated by high triglyceride levels. We did not find drug targets among the genes significantly dysregulated by RA (Supplementary Data 13).

**Tissue-specific effects**. Many traits and diseases manifest themselves only in certain tissues. For this reason, we performed tissue-specific revTWMR analyses using tissue-specific *trans*-eQTLs identified by the Genotype-Tissue Expression Project (GTEx)[47], which provides a unified view of genetic effects on gene expression across 49 human tissues. We tested the 51 previously identified significant trait → gene associations found in whole blood and detected three genes showing tissue-specific associations ($P_{revTWMR} < 0.05/51$, Supplementary Data 14). These include a negative effect of RA on *MYO1B* in the kidney cortex ($\alpha_{revTWMR} = -1.71$, $P_{revTWMR} = 2.4 \times 10^{-04}$), as well as a positive effect on *TRBV19* in the small intestine terminal ileum ($\alpha_{revTWMR} = 1.14$, $P_{revTWMR} = 1.7 \times 10^{-04}$) and in esophagus mucosa ($\alpha_{revTWMR} = 0.63$, $P_{revTWMR} = 5.7 \times 10^{-04}$). In addition, we observed a negative effect of HDL on *MYLIP* ($\alpha_{revTWMR} = -0.98$, $P_{revTWMR} = 5.9 \times 10^{-04}$) in the brain spinal cord (cervical c-1).

**Testing the reverse causal effects in mouse models**. RevTWMR pointed to 26 genes affected by TG levels. To experimentally validate these causal effects, we analyzed how the hepatic expression of these genes and TG levels were co-affected in the mouse BXD genetic reference panel, a set of inbred mice strains generated by crossing the C57BL/6 J and DBA/2 strains, upon switch to a high-fat diet (HFD)[48]. We hypothesized that if HFD-induced changes in TG correlate with HFD-induced changes in the expression of these genes, then diet-induced changes in TG might indeed be causal to changes in the expression of selected genes. Importantly, this relies on the assumption that there is a reasonable correlation between the expression of these genes in human blood and mice livers and that TG → gene expression mechanisms are conserved between the two species. Among the 19,942 genes tested in revTWMR, 10,841 had a detectable ortholog measured in the mouse samples. For each gene, we computed the Spearman correlation between HFD-induced expression fold change and the diet-induced TG differences as an indicator of genes perturbed by TG levels. Among the genes showing a significant ($P < 0.001$) correlation, we found an enrichment ($P_{Fisher} = 4.0 \times 10^{-04}$) of revTWMR genes ($P_{revTWMR} < 2.5 \times 10^{-06}$). Performing the same analysis on significant TWMR genes ($P_{TWMR} < 3 \times 10^{-06}$) did not yield an enrichment ($P_{Fisher} = 0.7$), confirming that correlations are mainly driven by the effect of TG on gene expression (Supplementary Data 15).

## Discussion

We presented a Mendelian randomization approach to study the impact of human phenotypes on the transcriptome. When calculating the reverse effect of phenotypes on gene expression it is important to note that findings from GWAS provide a measure of the genetic liability to develop a disease. In fact, using such genetic liability as exposure, the association with the gene expression does not necessarily reflect the consequence of the fully developed disease but might reflect a consequence of an early asymptomatic stage of the disease or a mere genetic predisposition[49]. Hence, these changes in the expression of revTWMR-implicated genes may occur before the disease manifest itself. As such, revTWMR results should not exclusively interpreted as markers of downstream mechanisms post-disease onset, but as potential early biomarkers.

Across the 46 genes identified by revTWMR, we observed a clear trend for functional relevance. Genes perturbed by complex diseases seem to confirm several previously reported associations between immune-related genes (*TRBV*) and RA[27]. In addition, revTWMR allowed gaining insight into the regulatory mechanisms controlling biological pathways, as illustrated with serum lipid levels. We observed that high HDL-cholesterol lowers the expression of genes involved in cholesterol biosynthesis (*FDFT1*) and cellular cholesterol uptake (*LDLR*), while it increases the expression of genes responsible for the degradation of *LDLR* (*MYLIP*) and cholesterol efflux (*ABCA1* and *ABCG1*). Together, this suggests that high HDL levels prevent intracellular cholesterol overload, which could explain its known cardioprotective effects[50]. However, TG levels, which were shown to independently increase risk of coronary artery disease (CAD)[51], impact the expression of the same genes in the opposite direction. Hence, high TG levels might increase CAD risk through an intracellular accumulation of cholesterol. The biological relevance of our findings is further supported by our drug target analysis, which found that four genes (*SREBF1*, *FDFT1*, *LDLR*, and *ABCA1*) whose expression was perturbed by serum lipid traits were targets of statins, a category of drugs aiming at regulating the very same traits. Lipids are major modulators of CAD risk[50,51] and established regulators of gene expression[52]. Hence, drugs targeting these downstream genes might modulate CAD risk, even though mediation analysis is warranted to support this hypothesis.

Combining results of DEG analysis and bidirectional TWMR allowed decomposing the observational correlation between whole blood gene expression and complex traits. This analysis showed that DEGs often reflect disease-induced changes in the transcriptome rather than disease-causing ones. Importantly, we observed that most of the correlation between gene expression and complex traits is due to confounders, which could partially be explained by age and sex being important determinants of both. The remaining correlation can almost entirely be explained by the trait-to-gene expression causal effects. Just like single SNPs, the individual expression of most genes has only a minute contribution to the phenotype, even if cumulatively their effect can be substantial. Diseases, however, represent a major burden for the organism, which can lead to drastic changes in the transcriptome program. In light of these considerations, one would expect that the correlation between a gene's expression level and a complex trait is reflecting disease status rather than an expression-to-trait link. Validating revTWMR requires large cohorts in which gene expression is measured before and after trait-modifying interventions. As conducting such studies in humans imposes serious logistic and ethical hurdles, we turned to mice studies to assess the impact of diet-induced changes in TG on gene expression and found that genes detected by TG-revTWMR are enriched among mouse orthologs whose HFD-induced changes in expression correlate with HFD-induced changes in TG.

Our approach has its limitations, which need to be considered when interpreting results. First, our results are mainly focused on gene expression levels in whole blood. This is primarily due to the reduced power resulting from the small sample sizes when conducting tissue-specific analyses. However, because gene regulation is tissue-specific and many diseases manifest themselves only in certain tissues, future possibilities to interrogate larger and more diverse tissue-specific *trans*-eQTL datasets could unravel causative disease-gene links for genes not differentially expressed in blood. This speculation is supported both by the fact that the effect sizes of the few tissue-specific associations we detected were more than fivefold larger than those estimated using whole blood data, as well as recent reports suggesting that *trans*-eQTLs are particularly cell type-specific[53].

Another caveat lies in the fact that differences in power makes it difficult to compare the results of TWMR and revTWMR. One of the most important determinants of statistical power for MR is the sample size available for the outcome, thus revTWMR is less powered, picking up mostly strong effects. Still, another factor influencing power is the number and strength of instruments. Hence, TWMR results will be more accurate once larger eQTL datasets become available, which will in turn increase the number of testable genes (currently 16 K). Finally, as with every MR approach, revTWMR is at risk of violating the MR assumptions. In particular, horizontal pleiotropy and indirect effects of the instruments on the exposures can substantially bias causal effect estimates. RevTWMR assumes that the top GWAS SNPs have a direct effect on the phenotype. In particular, correlated pleiotropy can lead to biased causal effect estimates and currently available methods that attempt to tackle such MR violations (e.g., CAUSE[54], LHC-MR[55], MR-APSS[56]) require genome-wide summary statistics, which is not yet available for transcripts in a large enough sample size. However, many SNPs show indirect or pleiotropic effects. We, therefore, mitigate the influence of these potential biases by excluding pleiotropic SNPs failing the heterogeneity test. Further gain in robustness should be obtained by integrating additional phenotypes as exposures through which instruments may act, as accounting for pleiotropy is a better approach than excluding violating instruments. Such a multi-phenotype revTWMR approach will be possible only once genome-wide *trans*-eQTLs summary statistics will become available.

A very exciting perspective is that revTWMR can theoretically be extended to other types of omics data, e.g., integrating methylomics data, as alterations in DNA methylation are more often the consequence rather than the cause of diseases[57]. One could apply the approach to protein levels (revPWMR) to gain further insights into the effects of complex traits on biomarkers but the sample size of proteomics datasets are currently too small.

In conclusion, our bidirectional analysis disentangles the causes and consequences of gene expression for complex traits and reveals that complex traits have a more pronounced impact on gene expression than the reverse. Therefore, studies comparing gene expression levels of diseased and healthy subjects may still point to useful biomarkers of disease predisposition or severity, but interventions that restore levels of the biomarker to normal levels will not necessarily be disease-modifying.

## Methods

### Reverse transcriptome-wide Mendelian randomization (revTWMR).
RevTWMR is a multi-instrument MR approach designed to estimate the causal effect of the phenotypes (exposure) on gene expression (outcome). For each gene, using an inverse-variance weighted method for summary statistics[16], we define the joint causal effect of the phenotypes on the outcome as

$$\hat{\alpha} = \left(\hat{\beta}' C^{-1} \hat{\beta}\right)^{-1} \left(\hat{\beta}' C^{-1} \hat{\gamma}\right) \quad (2)$$

Here $\beta$ is an *n*-vector that contains the standardized effect size of *n* independent SNPs on the phenotype, derived from GWAS. $\gamma$ is a vector of length *n* that contains the standardized effect size, in *trans*-, of each SNP on the gene expression. $C$ is the pair-wise LD matrix between the *n* SNPs.

As instrumental variables, we used independent ($r^2 < 0.01$) significant ($P_{GWAS} < 5 \times 10^{-08}$) SNPs chosen among the 10 K preselected trait-associated SNPs included in a *trans*-eQTL dataset from eQTLGen Consortium (31,684 whole blood samples). As we are using only strongly independent SNPs, we use the identity matrix to approximate $C$. The SNPs with larger effects on the outcome than on the exposure were removed, as these would indicate a violation of MR assumptions (likely reverse causality and/or confounding).

The variance of α can be calculated approximately by the Delta method

$$\text{var}(\hat{\alpha}) = \left(\frac{\partial \hat{\alpha}}{\partial \beta}\right)^2 * \text{var}\left(\hat{\beta}\right) + \left(\frac{\partial \hat{\alpha}}{\partial \gamma}\right)^2 * \text{var}(\hat{\gamma}) + \left(\frac{\partial \hat{\alpha}}{\partial \beta}\right) * \left(\frac{\partial \hat{\alpha}}{\partial \gamma}\right) * \text{cov}(\hat{\beta}, \hat{\gamma}) \quad (3)$$

where $\text{cov}(\beta, \gamma)$ is 0 if $\beta$ and $\gamma$ are estimated from independent samples. We defined the causal effect Z-statistic for gene $i$ as $\hat{\alpha}_i / \text{SE}(\hat{\alpha}_i)$, where $\text{SE}\left(\hat{\alpha}_i\right) = \sqrt{\text{var}(\hat{\alpha})_{i,i}}$.

We applied revTWMR across the human genome for a causal association between a set of 12 phenotypes and the expression levels of 19,942 genes using summary statistics from GWAS and eQTLs studies. The analysed traits include BMI, CD[17], EDU[20], FG[18], HDL-cholesterol, height, LDL-cholesterol, RA[19], SCZ[21], TC, TG, and WHRadjBMI. While for CD, EDU, FG, RA, SCZ, and WHRadjBMI summary statistics (estimated univariate effect size and standard error) originate from the most recent meta-analysis and were downloaded from the publicly available NIH Genome-wide Repository of Associations Between SNPs and Phenotypes (https://grasp.nhlbi.nih.gov/), for the other traits the GWAS were performed in UKBiobank and the summary statistics are from the Neale Lab (http://www.nealelab.is/uk-biobank/) (Supplementary Data 16). We only used SNPs on autosomal chromosomes and were available in the UK10K reference panel, which allowed estimating the LD among these SNPs and prune them. Strand ambiguous SNPs were removed.

**Heterogeneity test**. The validity of all MR approaches, such as revTWMR, relies on three assumptions. The third assumption (no pleiotropy) is crucial as MR causal estimates will be biased if the genetic variants (IVs) have pleiotropic effects[43]. Hence, revTWMR assumes that all genetic variants used as instrumental variables affect the outcome only through gene expression and not through independent biological pathways. To test for the presence of pleiotropy, we used Cochran's Q test[42,44]. In brief, we tested whether there is a significant difference between the revTWMR-effect of an instrument (i.e., $\alpha\beta_i$) and the estimated effect of that instrument on the gene expression ($\gamma_i$). We defined

$$d_i = \hat{\gamma}_i - \hat{\alpha}\hat{\beta}_i \quad (4)$$

and its variance as

$$\text{var}(d_i) = \text{var}(\hat{\gamma}_i) + (\beta_i)^2 * \text{var}(\hat{\alpha}) + \text{var}(\hat{\gamma}_i) * (\alpha)^2 + \text{var}\left(\hat{\beta}_i\right) * \text{var}(\hat{\alpha}) \quad (5)$$

Next, we tested the deviation of each SNP using the following test statistic

$$T_i = \frac{d_i^2}{\text{var}(d_i)} \sim \chi_1^2 \quad (6)$$

In case where $P < 1 \times 10^{-4}$, we removed the SNP with largest $|d_i|$ and then repeated the test.

**Transcriptome-wide Mendelian randomization (TWMR)**. In order to test the presence of a feedback loop of association, we ran TWMR[15] for all the significant revTWMR genes. To make TWMR and revTWMR results comparable, we ran a univariable TWMR where for each gene we estimated its total effect on the phenotype. The associations between the instrumental variables and the exposure (gene expression) and the outcome (complex traits) are estimated from the same studies used for revTWMR.

**Gene-based test**. To compare GWAS and revTWMR results, we performed gene-based test for association summary statistics using PASCAL[39]. PASCAL assesses the total contribution of all SNP within close physical proximity to a given gene by combining SNP association Z-statistics into gene-based $P$ values while accounting for local LD structure.

### Replication cohorts

#### EGCUT
*Study population*. The Estonian Genome Center, University of Tartu (EGCUT) cohort denotes the Estonian Biobank sample of more than 200,000 individuals or about 20% of the Estonian adult population. All Biobank participants have been genotyped and linked to electronic health records (EHR) of the Health Insurance Fund, national registries, and major hospitals. The EHR linkage captures the participants' medical history together with demographics, lifestyle information, and laboratory measurements; additional information is provided by self-completed questionnaires. Disease diagnoses are in the form of ICD-10 codes. RNA-seq data is available on 491 unrelated individuals. All Biobank participants have signed a broad informed consent to allow using their genetic and medical information for research purposes.

*Whole-blood-transcriptome analysis*. The preparation of RNA-seq data has been described in detail elsewhere[58]. RNA-seq reads were trimmed of adapters together with low-quality leading and trailing bases using Trimmomatic (version 0.36)[59]. Additional quality control was performed with FastQC (version 0.11.2). The final set of reads were mapped to a human genome reference version GRCh37.p13 using STAR (version 2.4.2a)[60]. Sample mix-ups were tested and corrected for using MixupMapper[61]. Principal component analysis on RNA-seq read counts revealed a batch of outlying samples which was uncovered to be due to a technical problem in library preparation—affected samples were discarded. Data were normalized using the weighted trimmed mean of *M* values[62] and used as log2-transformed counts per million. To account for (hidden) batch effects, the sequencing batch date together with the first gene expression principal components were used in all subsequent analyses.

## InChianti

*Study population.* The InCHIANTI study is a population-based sample that includes 298 individuals of age <65 years and 1155 individuals of age ≥65 years. The study design and protocol have been described in detail previously[63]. The data collection started in September 1998 and was completed in March 2000. The INRCA Ethical Committee approved the entire study protocol.

*Whole-blood-transcriptome analysis.* Peripheral blood specimens were collected from 712 individuals using the PAXgene tube technology to preserve levels of mRNA transcripts. RNA was extracted from peripheral blood samples using the PAXgene Blood mRNA kit (Qiagen, Crawley, UK) according to the manufacturer's instructions.

RNA was biotinylated and amplified using the Illumina® TotalPrep(tm) −96 RNA Amplification Kit and directly hybed with Human HT-12_v3 Expression BeadChips that include 48,803 probes. Image data were collected on an Illumina iScan and analysed using Illumina GenomeStudio software. These experiments were performed as per the manufacturer's instructions and as previously described[64]. Quality-control analysis of gene expression levels were previously described[65].

## SHIP-Trend

*Study population.* The Study of Health in Pomerania (SHIP-Trend) is a longitudinal population-based cohort study in West Pomerania, a region in the northeast of Germany, assessing the prevalence and incidence of common population-relevant diseases and their risk factors. Baseline examinations for SHIP-Trend were carried out between 2008 and 2012, comprising 4420 participants aged between 20 and 81 years. Study design and sampling methods were previously described[66]. The medical ethics committee of the University of Greifswald approved the study protocol, and oral and written informed consents were obtained from each of the study participants.

*Whole-blood-transcriptome analysis.* Blood sample collection, as well as RNA preparation, were described in detail elsewhere[67]. Briefly, whole-blood samples of a subset of SHIP-TREND were collected from the participants after overnight fasting (≥10 h) and stored in PAXgene Blood RNA Tubes (BD). Subsequently, RNA was prepared using the PAXgeneTM Blood miRNA Kit (QIAGEN, Hilden, Germany). The purity and concentration of RNA were determined using a NanoDrop ND-1000 UV-Vis Spectrophotometer (Thermo Scientific). To ensure a constantly high quality of the RNA preparations, all samples were analyzed using RNA 6000 Nano LabChips (Agilent Technologies, Germany) on a 2100 Bioanalyzer (Agilent Technologies, Germany) according to the manufacturer's instructions. Samples exhibiting an RNA integrity number (RIN) less than seven were excluded from further analysis. The Illumina TotalPrep-96 RNA Amplification Kit (Ambion, Darmstadt, Germany) was used for reverse transcription of 500 ng RNA into double-stranded (ds) cDNA and subsequent synthesis of biotin-UTP-labeled antisense-cRNA using this cDNA as the template. Finally, in total 3000 ng of cRNA were hybridized with a single array on the Illumina Human HT-12 v3 BeadChips, followed by washing and detection steps in accordance with the Illumina protocol. Processing of the SHIP-Trend RNA samples was performed at the Helmholtz Zentrum München. BeadChips were scanned using the Illumina Bead Array Reader. The Illumina software GenomeStudio V 2010.1 was used to read the generated raw data, for imputation of missing values and sample quality control. Subsequently, raw gene expression data were exported to the statistical environment R, version 2.14.2 (R Development Core Team 2011). Data were normalized using quantile normalization and log2-transformation using the lumi 2.8.0 package from the Bioconductor open-source software (http://www.bioconductor.org/). Finally, 991 samples were available for gene expression analysis. Technical covariates used in all statistical models included RNA amplification batch, RNA quality (RIN), and sample storage time. The SHIP-Trend expression dataset is available at GEO (Gene Expression Omnibus) public repository under the accession GSE 36382: 991 samples were available for analysis.

## Phenotype-gene expression correlation

To calculate the correlation between the phenotypes and the gene expression levels, we asked each cohort to run the following analysis. First, the inverse normal transformation was applied to phenotypes and gene expression. Next, transformed phenotypes were adjusted only for sex, age, and age[2], while gene expression was also corrected for other known relevant covariates. Finally, Pearson's correlation was calculated between the adjusted trait and the adjusted expression. Finally, correlations from single cohorts were combined using inverse-variance meta-analysis, where weights are proportional to the squared standard error of the correlation estimates, as implemented in METAL[68].

## Observed and true correlation between gene expression and traits

The correlation between the effects estimated by revTWMR ($\alpha_{revTWMR}$) and the observational correlation ($corr(E, T)$) measured in the individual data from EGCUT, InChianti, and SHIP-Trend was calculated using Pearson's correlation. As such correlation does not consider the error of the estimations, for the significant correlations we used the linear errors-in-variables models to compute the potential

true correlation using the following equation

$$corr_{obs} = corr_{true} * \sqrt{1 - \frac{\sum_{j=1}^{N_{Genes}} SE(\alpha_{revTWMR})^2}{\sum_{j=1}^{N_{Genes}} \alpha_{revTWMR}^2}} * \sqrt{1 - \frac{\sum_{j=1}^{N_{Genes}} SE(corr(E, T))^2}{\sum_{j=1}^{N_{Genes}} corr(E, T)^2}}$$

(7)

## Proportion of observational correlation explained by bidirectional causal effects

Let $E$ and $T$ denote the gene expression and the trait, respectively. In addition, there may exist a confounding factor $U$ causally impacting both of them. We can express $E$ and $T$ as:

$$T = \alpha_{TWMR} * E + q_T * U + \varepsilon_T$$

(8)

And

$$E = \alpha_{revTWMR} * T + q_E * U + \varepsilon_E$$

(9)

where $\alpha_{TWMR}$ and $\alpha_{revTWMR}$ are the causal effects of $E$ on $T$ and of $T$ on $E$ estimated by TWMR and revTWMR respectively; $q_T$ and $q_E$ are the causal effects of the confounders on $T$ and $E$; and $\varepsilon_T \sim N(0, \sigma_T)$ and $\varepsilon_E \sim N(0, \sigma_E)$ represent uncorrelated errors. More specifically, $\varepsilon_T$, $\varepsilon_E$, and $U$ are all independent of each other, because all dependence between $T$ and $E$ are due to bidirectional causal effects and the confounder $U$, the residual noises are independent of each other and of the confounder.

For simplicity, we assume that $E$, $T$, and $U$ have zero mean and unit variance, so that the correlation between $E$ and $T$ can be expressed as

$$corr(E, T) = cov(E, T) = E(E * T) = \alpha_{TWMR} + \alpha_{revTWMR} - \alpha_{TWMR} * \alpha_{revTWMR} * E(E * T) + q_T * q_E$$

(10)

Equivalently,

$$corr(E, T) = \frac{\alpha_{TWMR} + \alpha_{revTWMR} + q_T * q_E}{1 + \alpha_{TWMR} * \alpha_{revTWMR}}$$

(11)

As we know the correlation, the bidirectional causal effects estimated by TWMR and revTWMR, we can estimate the contribution of the confounders ($q_T * q_E$) to the observed correlation. Since the magnitude of $\alpha_{TWMR} * \alpha_{revTWMR}$ is negligible, we replaced the denominator with 1.

To avoid the recursive equations expressing the forward and reverse causal effects of E on T, we can substitute $T$ into the equation for $E$ and obtain

$$E = \alpha_{revTWMR} * (\alpha_{TWMR} * E + q_T * U + \varepsilon_T) + q_E * U + \varepsilon_E$$
$$E = \alpha_{reTWMR} * \alpha_{TWMR} * E + (\alpha_{revTWMR} * q_T + q_E) * U + \alpha_{revTWMR} * \varepsilon_T + \varepsilon_E$$
$$(1 - \alpha_{revTWMR} * \alpha_{TWMR}) * E = (\alpha_{revTWMR} * q_T + q_E) * U + \alpha_{revTWMR} * \varepsilon_T + \varepsilon_E$$
$$E = \frac{(\alpha_{revTWMR} * q_T + q_E) * U + \alpha_{revTWMR} * \varepsilon_T + \varepsilon_E}{1 - \alpha_{revTWMR} * \alpha_{TWMR}}$$

(12)

Similarly for $T$

$$T = \alpha_{TWMR} * (\alpha_{revTWMR} * T + q_E * U + \varepsilon_E) + q_T * U + \varepsilon_T$$
$$T = \alpha_{TWMR} * \alpha_{revTWMR} * T + (\alpha_{TWMR} * q_E + q_T) * U + \alpha_{TWMR} * \varepsilon_E + \varepsilon_T$$
$$(1 - \alpha_{TWMR} * \alpha_{revTWMR}) * T = (\alpha_{TWMR} * q_E + q_T) * U + \alpha_{TWMR} * \varepsilon_E + \varepsilon_T$$
$$T = \frac{(\alpha_{TWMR} * q_E + q_T) * U + \alpha_{TWMR} * \varepsilon_E + \varepsilon_T}{1 - \alpha_{TWMR} * \alpha_{revTWMR}}$$

(13)

## GWAS hits *trans*-eQTL mapping in GTEx

Genotypes and gene expression quantifications from the GTEx project v8 dataset[47] were obtained via dbGaP accession number phs000424.v8.p1. This includes genotypes of 838 subjects, 85.3% of European American origin, 12.3% African American, and 1.4% Asian American. The phased version of the genotype files was used and the genotypes for 1078 out of 1093 GWAS hits used as instrument variables in revTWMR were retrieved, matching for chromosome, position, and reference/alternative allele, after conversion to GRCh38 coordinates using the UCSC liftOver tool[69]. Gene expression quantification (TPM values) from RNA-seq experiments across 49 tissues (for which genotype data is also available for ≥70 individuals) processed and provided by the GTEx project v8 were also downloaded. These gene expression quantifications had been mapped to Gencode v26[70] gene annotations on GRCh38 and normalized by TMM between samples (as implemented in edgeR), and inverse normal transform across samples. Moreover, only genes passing an expression threshold of >0.1 TPM in ≥20% samples and ≥6 reads in ≥20% samples had been retained. The association between each of the 2177 GWAS hits genotyped in GTEx v8 and each gene expression (20,315 to 35,007 genes per tissue, all gene types) across 49 tissues of the GTEx v8 was computed using QTLtools v1.3.1 trans function[71]. This consists of more than 2 billion association tests performed. For this, the–nominal option for calculating nominal $p$ values was used, as well as the–normal option, to enforce the gene expression phenotypes to match normal distributions $N(0,1)$. To include all associations, no *cis* window filtering was applied. Moreover, covariates provided by GTEx v8 for each tissue were regressed out of each expression matrix to account for potential confounding factors, by using the–covariate option on QTLtools. These included 15 to 60 PEER factors (depending on tissue sample size)[72], five genotype PCA PCs as well as information

about the sequencing platform, PCR usage, and the sex of the samples provided by GTEx v8.

**Analysis of mouse data**. We used blood triglyceride and liver gene expression in a panel of BXD mice that were fed a chow or high-fat diet (CD and HFD[48]). The study involves 52 strains, with five mice per strain per condition. Male mice were switched to an HFD diet at 8 weeks of age, subjected to extensive cardiometabolic phenotyping, and finally sacrificed at 29 weeks of age after an overnight fast. The blood and liver collection were performed simultaneously during tissue collection. Microarray data and triglyceride measurements in the two diets are available for a subset of 34 strains. Phenotype data, including blood triglyceride measurement, are deposited in the Mouse Phenome Database (https://phenome.jax.org/projects/Auwerx1) and the raw gene expression data in the Gene Expression Omnibus (https://www.ncbi.nlm.nih.gov/geo/query/acc.cgi?acc=GSE60149). To calculate diet-induced TG change, we subtracted the strain average on CD from that on HFD and converted the resulting difference into a z-score. We normalized the Affymetrix Mouse Gene 1.0 ST Array data using the Affymetrix Power Tools software version 1.20.5 with GC correction (GCCN) and space transformation (SST). We removed the lowest quartile of genes based on average expression in all samples. For each BXD strain, we calculated strain-level HFD-induced fold change as the difference in the expression on HFD minus that of CD and then converted these values to z-scores. We then performed Spearman's correlation for each gene's HFD-induced fold change and the TG diet-induced differences. We used the biomaRt R package version 2.42.1[73] to convert between mouse and human gene Ensembl IDs.

**Reporting Summary**. Further information on research design is available in the Nature Research Reporting Summary linked to this article.

## Data availability

The *trans*-eQTLs data used in this study are available on the eQLTGen Consortium website [https://www.eqtlgen.org/trans-eqtls.html]. The GWAS data are available in the NIH Genome-wide Repository of Associations Between SNPs and Phenotypes [https://grasp.nhlbi.nih.gov/] and in the Neal Lab website [http://www.nealelab.is/uk-biobank/]. For the mouse data, phenotype data, including blood triglyceride measurement, are deposited in the Mouse Phenome Database [https://phenome.jax.org/projects/Auwerx1] and the raw gene expression data in the Gene Expression Omnibus [https://www.ncbi.nlm.nih.gov/geo/query/acc.cgi?acc=GSE60149]. Source data are provided with this paper.

## Code availability

R-code for performing revTWMR analyses is available at https://github.com/eleporcu/revTWMR https://doi.org/10.5281/zenodo.5119244.

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

## Acknowledgements

This work was supported by grants from the Swiss National Science Foundation (310030-189147, 32473B-166450, 32003B-173092 to Z.K. and 31003A_182632 to A.R.) and Horizon2020 Twinning projects (ePerMed 692145 to A.R.). SHIP-Trend is part of the Community Medicine Research net of the University of Greifswald, Germany, which is funded by the Federal Ministry of Education and Research (grants no. 01ZZ9603, 01ZZ0103, and 01ZZ0403), the Ministry of Cultural Affairs as well as the Social Ministry of the Federal State of Mecklenburg-West Pomerania, and the network "Greifswald Approach to Individualized Medicine (GANI_MED)" funded by the Federal Ministry of Education and Research (grant 03IS2061A). The University of Greifswald is a member of the Caché Campus program of the InterSystems GmbH. Research on BXD mice was supported by the Ecole Polytechnique Fédérale de Lausanne (EPFL), European Research Council (ERCAdG-787702), and Swiss National Science Foundation (SNSF 310030B160318). We would like to thank Liza Darrous and Ninon Mounier for their valuable feedback and comments on this manuscript.

## Author contributions

E.P. and Z.K. conceived and designed the study; E.P. and Z.K. contributed to the mathematical derivations of the research; E.P. performed statistical analyses; M.C.S. carried out drug target analyses; Z.K. supervised drug target analyses; K.L. has performed initial comparisons between gene expression-trait correlation and TWMR effects in the EGCUT cohort; C.A. and F.A.S. contributed with the biological interpretation of the results; E.P., A.R. and Z.K. drafted the manuscript; M.C.S. and C.A. contributed to the writing of specific sections; C.A. and F.A.S. revised the manuscript; K.L. performed statistical analyses on EGCUT cohort; A.M. oversaw the analysis in EGCUT; A.R.W. performed statistical analyses on InChianti cohort; S.B., T.T. and T.F. oversaw the analysis in InChianti; A.W. performed statistical analyses on SHIP-Trend cohort; U.V. and M.N. contributed to the data collection, quality control, and study design of SHIP-Trend; A.T. oversaw the analysis in SHIP-Trend; M.S.B.S. performed the analysis in mice data; D.M.R. performed *trans*-eQTLs analyses in GTEx dataset; O.D. designed and supervised *trans*-eQTLs analyses on GTEx dataset. All authors read the paper and contributed to its final form.

## Competing interests

The authors declare no competing interests.
