## [Peer Review File · Nature Communications]

Reviewers' Comments:

Reviewer #1:

Remarks to the Author:

The authors have addressed a large proportion of my comments. I like some of the new results they added, especially the one about high-fat induced mice. Still, I have a couple of remaining concerns:

(1) I think it would be helpful to show revTWMR results in the form of scatter plots of effect sizes for a few trait-gene pairs. One such figure is shown for the STX1B gene, and it's visually clear that the pleiotropic SNP causes the problem and should be removed. For the genes that have passed revTWMR (and the filter), i.e. the genes in Table S1, it would be good to show some similar plots. It would add confidence of the results. My concern is that some of the results may be driven by just a small number of SNPs, and the heterogeneity filter does not recognize that. This is related to the point of "correlated pleiotropy", which is difficult to account for. Also, as a general comment, this kind of plots for one or few examples would give readers a better sense of how revTWMR works, so it would be good to add anyway.

(2) About the finding that some genes presumably affected by HDL, but are actually drug targets: the authors cited a perspective. However, it answers a different question, about association of PRS of a trait with a gene. Of course, in that scenario, some of the associations could be driven by true causal genes. But here, the test is explicit about reverse causality, so I still find the results a bit puzzling. Perhaps it would be useful to show a few plots as suggested in comment (1).

Reviewer #2:

Remarks to the Author:

I thank the authors for their response. They have addressed all my major comments.

Minor points.

1. Although I agree that it is not possible/feasible to address violations to the correlated pleiotropy assumption in their paper, the authors should mention in the manuscript text that this assumption is being made (and to cite the CAUSE paper as one approach to account for violations to this assumption).

Reviewer #3:

Remarks to the Author:

Thank you for the opportunity to review this manuscript. Porcu and colleagues have built upon their previous research in pioneering methodology regarding transcriptome-wide Mendelian randomization with a novel approach to explore genetically predicted bi-directional effects between gene expression and complex traits/disease endpoints. Their approach has highlighted several noteworthy findings which makes a compelling case for the application of this method to other large-scale molecular datasets e.g. the GoDMC consortium for DNA methylation, and recent metabolites GWAS as the authors make the case for their response to a previous reviewer comment.

The authors have clearly already put a huge amount of effort into addressing comments from the first round of review which I was not involved in (particularly the added animal work). I believe this initial round of reviews has really helped refine the manuscript which in my opinion will be of

interest to the readership of Nature Communications. I therefore have only provided a few minor comments to add to review process and would like to congratulate the authors on their excellent paper.

Minor comments:

- Could the authors make it clearer where genetic instruments for complex traits/diseases were obtained from in their study (perhaps in a supplementary table)? It seems that for certain diseases these were obtained from summary statistics provided by studies (CD, EDU, FG, RA and SCZ) - whereas the remainder were derived from the UK Biobank or obtained from the NIH Grasp website. I couldn't find this in the supplementary materials but I think readers would like to see exactly what instruments were used for these exposures in the revTWMR.
- The finding that RA provided 15 hits for revTWMR is quite intriguing - particularly as the RA instruments are from smaller GWAS samples compared to some other exposures (I would have suspected BMI as having the most hits a priori). Do the authors have any hypotheses about why this may be the case? Could tissue-specificity be relevant here i.e. whole blood is capturing effects for an autoimmune system disease more strongly than exposures which could be more tissue-dependent (e.g. BMI effects may be more confined to brain/adipose/muscle tissues?). That said, triglycerides provided plenty of hits and we might expect those to be more confined to liver tissue.
- At the start of the discussion to author make the important point that revTWMR-implicated genes are based on estimates related to disease liability (rather than disease itself). This is a crucial point to clarify in the updated manuscript which is great to see. They then claim that results of revTWMR may be of interest as indications of potential early biomarkers (which I agree with) but also drug targets (which I'm skeptical of). I would suggest removing this point about drug targets and move it further down in the discussion in future work where the authors suggest their methodology could be extended to circulating proteins.
- In the limitations section of the discussion the authors claims that their heterogeneity filtering can 'protect our results from potential biases'. I think this should be toned down to expressing that this sensitivity analysis can 'help mitigate the influence of these potential biases' rather than completely protect against them.

Reviewer #1 (Remarks to the Author):

The authors have addressed a large proportion of my comments. I like some of the new results they added, especially the one about high-fat induced mice. Still, I have a couple of remaining concerns:

We thank the reviewer for his/her praise. We will do our best to clarify the parts that are still unclear.

(1) I think it would be helpful to show revTWMMR results in the form of scatter plots of effect sizes for a few trait-gene pairs. One such figure is shown for the STX1B gene, and it's visually clear that the pleiotropic SNP causes the problem and should be removed. For the genes that have passed revTWMMR (and the filter), i.e. the genes in Table S1, it would be good to show some similar plots. It would add confidence of the results. My concern is that some of the results may be driven by just a small number of SNPs, and the heterogeneity filter does not recognize that. This is related to the point of "correlated pleiotropy", which is difficult to account for. Also, as a general comment, this kind of plots for one or few examples would give readers a better sense of how revTWMMR works, so it would be good to add anyway.

We agree with the reviewer that such plots would add “visual confidence” to the results. We added the Supplementary Figure 1 where we included the scatter plots for all the genes mentioned in the results section. These plots indicate that (i) the regression slopes are not driven by a few outlying SNPs and (ii) it is also visible that correlated pleiotropy should play a minor role, otherwise we would observe two distinct clusters of SNPs within these plots, one with a slope due to the causal effect and another with a different slope (representing the ratio of the causal effect of a confounder on the exposure and outcome).

(2) About the finding that some genes presumably affected by HDL, but are actually drug targets: the authors cited a perspective. However, it answers a different question, about association of PRS of a trait with a gene. Of course, in that scenario, some of the associations could be driven by true causal genes. But here, the test is explicit about reverse causality, so I still find the results a bit puzzling. Perhaps it would be useful to show a few plots as suggested in comment (1).

The plots for *FDFT1*, *ABCG1* and *ABCA1* are now included in Supplementary Figure 1. We agree with the reviewer that it is puzzling how a gene indicated by revTWMMR could be a drug target. Answering this requires more data (*cis/trans* eQTL data, longitudinal studies) and more tailored methods, which we will pursue as a future research direction. For HDL, in particular, many key genes do not have (direct) (e)QTLs, hence *cannot be tested by TWMMR*. However, the expression of these genes seems to be driven by the polygenic predisposition to high HDL levels [see Discussion] and such gene expressions might mediate the genetic predisposition.

Reviewer #2 (Remarks to the Author):

I thank the authors for their response. They have addressed all my major comments.

Minor points.

1. Although I agree that it is not possible/feasible to address violations to the correlated pleiotropy assumption in their paper, the authors should mention in the manuscript text that this assumption is being made (and to cite the CAUSE paper as one approach to account for violations to this assumption).

Following this comment, we now mention in the Discussion section correlated pleiotropy as a potential limitation and cite CAUSE and two other preprint papers that aim to tackle correlated pleiotropy and explain why they are not applicable currently due to data availability .

“In particular, correlated pleiotropy can lead to biased causal effect estimates and currently available methods that attempt to tackle such MR violations (e.g. CAUSE [54], LHC-MR [55], MR-APSS [56]) require genome-wide summary statistics, which is not yet available for transcripts in large-enough sample size.”

Reviewer #3 (Remarks to the Author):

Thank you for the opportunity to review this manuscript. Porcu and colleagues have built upon their previous research in pioneering methodology regarding transcriptome-wide Mendelian randomization with a novel approach to explore genetically predicted bi-directional effects between gene expression and complex traits/disease endpoints. Their approach has highlighted several noteworthy findings which makes a compelling case for the application of this method to other large-scale molecular datasets e.g. the GoDMC consortium for DNA methylation, and recent metabolites GWAS as the authors make the case for their response to a previous reviewer comment.

The authors have clearly already put a huge amount of effort into addressing comments from the first round of review which I was not involved in (particularly the added animal work). I believe this initial round of reviews has really helped refine the manuscript which in my opinion will be of interest to the readership of Nature Communications. I therefore have only provided a few minor comments to add to review process and would like to congratulate the authors on their excellent paper.

We thank the reviewer for his/her praise and recognizing the value of our work.

Minor comments:

- Could the authors make it clearer where genetic instruments for complex traits/diseases were obtained from in their study (perhaps in a supplementary table)? It seems that for

certain diseases these were obtained from summary statistics provided by studies (CD, EDU, FG, RA and SCZ) - whereas the remainder were derived from the UK Biobank or obtained from the NIH Grasp website. I couldn't find this in the supplementary materials but I think readers would like to see exactly what instruments were used for these exposures in the revTWMR.

We apologize for the lack of clarity. We added Supplementary Table 16 where for each phenotype we indicated the reference paper and the link to download the summary statistics.

- The finding that RA provided 15 hits for revTWMR is quite intriguing - particularly as the RA instruments are from smaller GWAS samples compared to some other exposures (I would have suspected BMI as having the most hits a priori). Do the authors have any hypotheses about why this may be the case? Could tissue-specificity be relevant here i.e. whole blood is capturing effects for an autoimmune system disease more strongly than exposures which could be more tissue-dependent (e.g. BMI effects may be more confined to brain/adipose/muscle tissues?). That said, triglycerides provided plenty of hits and we might expect those to be more confined to liver tissue.

We agree with the reviewer that here tissue specificity plays a crucial role. However, the lack of *trans* eQTL data in other tissues makes any hypothesis difficult to be explored. A recent work from METABRAIN consortium analyzed *cis* and *trans* eQTLs in several regions of the brain showing that while *cis* eQTLs are shared between brain and whole blood, *trans* eQTLs seem to be tissue specific (<https://doi.org/10.1101/2021.03.01.433439>). While these results could explain why we did not see any significant gene affected as their primary “tissue of action” is not transcriptionally correlated to whole blood (e.g. BMI, for which it is brain), it is possible that there is a greater overlap between liver and whole blood *trans* eQTLs, explaining the results found for lipids. A possible alternative explanation is that triglycerides levels were measured in the blood, this tissue still may play some role beyond liver.

-- At the start of the discussion to author make the important point that revTWMR-implicated genes are based on estimates related to disease liability (rather than disease itself). This is a crucial point to clarify in the updated manuscript which is great to see. They then claim that results of revTWMR may be of interest as indications of potential early biomarkers (which I agree with) but also drug targets (which I'm skeptical of). I would suggest removing this point about drug targets and move it further down in the discussion in future work where the authors suggest their methodology could be extended to circulating proteins.

We removed the point about drug targets from the first part of the discussion. In the current form of the manuscript we discussed the results we obtained and we recognized that further analysis are needed to validate our hypothesis:

“The biological relevance of our findings is further supported by our drug target analysis, which found that four genes (*SREBF1*, *FDFT1*, *LDLR* and *ABCA1*) whose

expression was perturbed by serum lipid traits were targets of statins, a category of drugs aiming at regulating the very same traits. Lipids are major modulators of CAD risk [49, 50] and established regulators of gene expression [51]. Hence, drugs targeting these downstream genes might modulate CAD risk, even though mediation analysis is warranted to support this hypothesis.”

- In the limitations section of the discussion the authors claims that their heterogeneity filtering can 'protect our results from potential biases'. I think this should be toned down to expressing that this sensitivity analysis can 'help mitigate the influence of these potential biases' rather than completely protect against them.

We modified the text as suggested

“We therefore mitigate the influence of these potential biases by excluding pleiotropic SNPs failing the heterogeneity test.”

Reviewers' Comments:

Reviewer #1:

Remarks to the Author:

The authors have addressed all my concerns. I'd recommend for publication.

Reviewer #3:

Remarks to the Author:

Thank you for addressing my comments. Congratulations on an excellent paper.

Tom Richardson